# Experimental observation of quantum mechanical fluorine tunnelling

Carsten Müller[1], Frederik Bader[2], Frenio A. Redeker [3], Lawrence Conrad[2], Helmut Beckers[1], Beate Paulus [2] ✉, Sebastian Riedel [1] ✉ & Jean Christophe Tremblay [4] ✉

Quantum mechanical tunnelling occurs when a molecule transforms between two states separated by a finite energy barrier that cannot be overcome thermally. To date, it has been observed for elements up to oxygen. Efforts to go one element further are hindered by the strong bonds formed by fluorine with other elements, which suppress tunnelling. In this work, laser ablation is used to create fluorine-only species and trap weakly-bound polyfluorides in a neon matrix at cryogenic temperatures. Spectroscopic investigations reveal a temperature-dependent doublet-splitting, providing experimental evidence for heavy-atom quantum mechanical tunnelling. Theoretical modelling attributes the signal to tunnelling of the central fluorine atom in a quasi-linear $[F_2 \cdots F \cdots F_2]^-$ complex through a rotational barrier caused by steric hindrance and electronic effects in the neon matrix. The present study offers new insights into chemical interactions in polyfluorides and, more generally, of quantum phenomena in confined environments.

Polyhalogen anions are extensively studied due to their potential applications in a broad variety of fields including redox-flow batteries[1,2], separation of metals[3–6], organic and inorganic synthesis[7–9], as well as safe and easy to handle halogen storage material[10]. The stability of the homoatomic polyhalogen anions increases significantly going from the lightest halogen fluorine (F) to the heaviest non-radioactive halogen iodine (I). Symmetric $[X_3]^-$ (X = F, Cl, Br, I) is characterized by a 3-center 4-electron (3c-4e) bond, which for asymmetric $[X_3]^-$ continuously transits to an interaction of a positive electrostatic potential that exists at both ends along the bond axis (σ-hole) of the dihalogen molecule ($X_2$) with the negative electrostatic potential of the halide $X^-$. For larger polyhalogen anions, 3c-4e bonds are of minor importance and the structures can be generally described in terms of σ-hole interactions between the units $X^-$, $X_2$, and $[X_3]^-$. The positive electrostatic potential is highest for the σ-hole of $I_2$ and decreases towards $F_2$ where it is weakly positive[11]. The large σ-hole of $I_2$ leads to a great variety of known polyiodine anions and anionic polyiodine networks[12]. Fewer structures are known for bromine and for chlorine[13], while for fluorine only $[F_3]^{-[14–20]}$ and $[F_5]^{-[21,22]}$ have been

observed and in contrast to I, Br, and Cl none of the polyfluorine anions have been isolated under ambient conditions[13].

The pentafluorine anion, $[F_5]^-$, is a very weakly bound complex, rendering its experimental characterization challenging. Contrary to the pentachlorine anion, $[Cl_5]^-$, which exists as bulk material and takes a hockey stick like structure[23], the existence of the pentafluorine anion remains the subject of debates. The first fragmental evidence of the formation of an $[F_5]^-$ ion was reported in 2000 by Artau et al., who observed a weak signal of m/z=95 in mass spectra while investigating the bond dissociation energies of $[F_3]^-$ (m/z=57) in the gas phase[17]. Ten years later the structure of the $[F_5]^-$ ion was investigated computationally at the CCSD(T)/aug-cc-pVTZ level of theory[18], which predicted a bent structure shaped like a hockey stick, consisting of an $[F_3]^-$ moiety weakly bound to an $F_2$ unit and transforming according to the $C_s$ point group. The IR bands associated with the $F_2$ and antisymmetric $[F_3]^-$ stretching vibrations were predicted at 805 and 528 cm$^{-1}$, respectively[18].

Another five years later, IR spectra obtained after co-deposition of laser ablated metals with $F_2$ in solid neon matrices at 5 K yielded the expected IR signals corresponding to metal fluorides and the free $[F_3]^-$

---

[1]Institute for Chemistry and Biochemistry, Freie Universität Berlin, Fabeckstrasse 34/36, Berlin, Germany. [2]Institute for Chemistry and Biochemistry, Freie Universität Berlin, Arnimallee 22, Berlin, Germany. [3]Department of Chemistry, Georgetown University, Washington, DC, USA. [4]CNRS-Université de Lorraine, LPCT, 1 Bd Arago, Metz, France. ✉e-mail: b.paulus@fu-berlin.de; s.riedel@fu-berlin.de; jean-christophe.tremblay@univ-lorraine.fr

anion (525 cm$^{-1}$). Two further bands at 850 and 1805 cm$^{-1}$ were observed experimentally[21,22] which, contrary to the other bands, disappeared after irradiation with red LED light. These new bands were tentatively assigned to the antisymmetric stretching vibration, and a combination band associated with the two $F_2$ units in a hypothetic V-shaped $[F_5]^-$ ion transforming according to the $C_{2v}$ point group. That assignment, however, is in contradiction with earlier computational results, which predicted the V-shaped $[F_5]^-$ ion to be a transition state[18]. The assignment of the observed 850 cm$^{-1}$ band to a hockey stick shaped $[F_5]^-$ ion does not match the calculated $F_2$ stretching frequency (805 cm$^{-1}$). The calculations also predict the presence of a strong antisymmetric stretching vibration at 528 cm$^{-1}$ attributed to its $[F_3]^-$ moiety but not observed experimentally. To resolve these contradictions it was hypothesized that only the V-shaped structure of $[F_5]^-$ might be stabilized by neighbouring cationic species in the neon matrix[21] – an hypothesis that could not been proven up to date.

Alternatively, the observations could be explained by quantum mechanical tunnelling (QMT). The importance of QMT to understanding chemical reactivity is often not recognized[24]. While there are many examples of heavy-atom QMT involving carbon atoms[25–27], there exist only a few examples for nitrogen[28,29] and oxygen[30–32]. Recently, a hypothetical double-germanium based fluoride receptor[33], as well as the pseudorotation in the isoelectronic series XeF$_6$, IF$_6^-$, and TeF$_6^{2-}$[34], have been computationally predicted to rearrange via heavy-atom QMT with fluorine as the tunnelling-determining atom – that is, the atom with the strongest impact on the tunnelling reaction[35]. Experimentally, QMT was also found to enhance the reactivity of fluorine with *para*-hydrogen[36] and its role was discussed in the context of matrix-isolation investigations of nitrosyl halides isomerizations[37].

Here, we present the first experimental and theoretical evidence for heavy-atom QMT of a tunnelling-determining fluorine atom that takes place in a polyfluoride species. This provides important insights into genuine fluorine-specific interactions in polyfluoride compounds and thereby reaches the so-called fluorine wall for tunnelling[38]. We argue that such confinement-induced tunnelling could become important for controlling and optimizing the rate of chemical reactions.

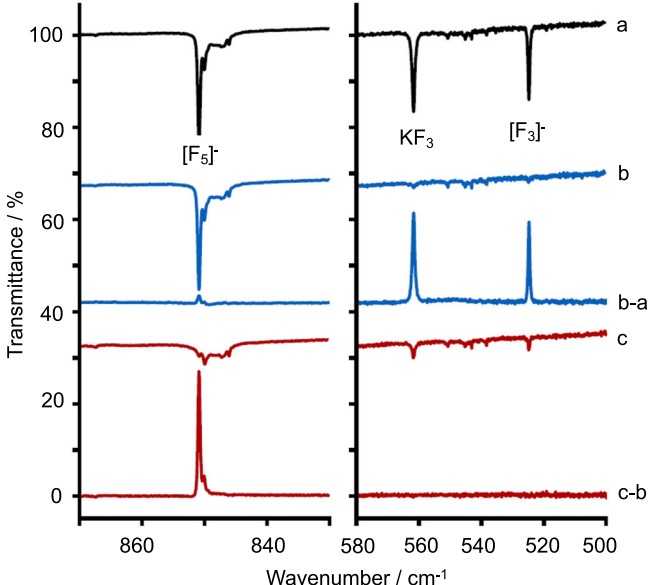

**Fig. 1 | Ne-matrix FTIR spectra obtained after co-deposition of potassium fluoride with 1% $F_2$ in solid neon at 5K. a** After 4 h deposition (black), **b** after 5 min irradiation with UV light at $\lambda$ = 273 nm (blue), c) after 5 min irradiation with red light at $\lambda$ = 730 nm (red). The difference spectra are shown in panels (**b**−**a** and **c**−**b**) in blue and red, respectively. Bands pointing up belong to species that are depleted whereas bands pointing down belong to species that are formed. Source data are provided as a Source Data file.

## Results

### Matrix-Isolation Experiments

Figure 1 shows Ne-matrix infrared spectra in the regions from 500 to 580 cm$^{-1}$ and from 830 to 870 cm$^{-1}$ that were obtained after co-deposition of laser-ablated potassium fluoride with 1% $F_2$ in solid neon at 5K. The spectrum recorded immediately after deposition (Fig. 1a) shows three bands in those regions at 525, 562, and 851 cm$^{-1}$, which are assigned to the antisymmetric stretching vibration of free $[F_3]^-$, the antisymmetric stretching vibration of the $[F_3]^-$ unit in the ion pair K$^+[F_3]^-$, and an antisymmetric stretching vibration in $[F_5]^-$, respectively. Upon subsequent irradiation with UV light ($\lambda$ = 273 nm) the two bands that belong to $[F_3]^-$ species are depleted (525 and 562 cm$^{-1}$, as can be seen by comparing Fig. 1b and b−a). This confirms that the signal at 851 cm$^{-1}$ belongs to a different chemical species.

Upon irradiation with red light ($\lambda$ = 730 nm) only the band attributed to $[F_5]^-$ is depleted ( ~851 cm$^{-1}$, see Fig. 1c and c−b), while small amounts of $[F_3]^-$ and K$^+[F_3]^-$ are formed. The formation of the latter is likely due to slight heating of the deposit during irradiation which leads to a reaction of F$^-$ or KF with $F_2$ upon reorientation inside a shared cavity. The $[F_5]^-$ ion does not have an IR active band in the antisymmetric $[F_3]^-$ stretching region, as can be clearly seen in Fig. 1b which shows no remaining bands in the region from 500 to 580 cm$^{-1}$ after depletion of $[F_3]^-$ and K$^+[F_3]^-$. The presence of a theoretically predicted strong antisymmetric stretching vibration associated with the $[F_3]^-$ moiety in the $[F_5]^-$ anion can therefore be ruled out. As will be argued below, this observation can be explained with symmetry selection rules: as the stabilization and compression inside the matrix increases, the symmetry of $[F_5]^-$ changes from planar $C_{2v}$ to quasi-linear $D_{\infty h}$. Thus, the fundamental stretching vibration observed in the IR spectrum of $[F_5]^-$ (Fig. 1b and c−b) can be interpreted as the displacement of the central F$^-$ ion between two polarized $[F_2]$ moieties.

Figure 2 shows Ne-matrix FTIR spectra in the region between 847 and 854 cm$^{-1}$ upon annealing. The IR spectra clearly show that the band at 851 cm$^{-1}$ consists in fact of two signals, one at 850.9 cm$^{-1}$ and a weaker one at 850.1 cm$^{-1}$. It also shows that the intensity of the band at 850.1 cm$^{-1}$ increases relative to the band at 850.9 cm$^{-1}$ if the deposit is exposed to the weak IR light source of the spectrometer for several hours (Fig. 2, signal b) and increases even further upon increasing the temperature to 10K (Fig. 2, signal c). This change of intensity is observed to be reversible. Both the splitting of the band itself and the fact that the relative intensity of the weaker band reversibly increases when thermal energy is provided are in support of the presence of a vibrationally excited state at a very low energy. Converting the intensity ratio at 10K using a Boltzmann distribution yields an estimate for the energy splitting of ~3.87 cm$^{-1}$. This is indicative of a tunnelling behaviour rather than to the presence of matrix sites. Matrix sites give rise to bands that are shifted with respect to the main band due to different molecular orientations within their cavity. These typically show the opposite thermal behavior and merge irreversibly with the main band upon annealing, because softening of the matrix leads to reorientation of these molecules. Hence, we can exclude that the weak band at 850.1 cm$^{-1}$ in Fig. 2 originates from matrix sites.

### Gas phase models

The isolated pentafluoride anion has a very flexible structure and its potential energy landscape presents multiple local energy minima connected by low energy barriers. Representative structures and their relative energies are shown in Fig. 3. First principles electronic structure calculations[39] were used to investigate the low-energy structures of $[F_5]^-$ (see Supplementary Materials (SM) for computational details). The global energy minimum has an asymmetrical $C_s$ structure shaped like a hockey stick connecting an $[F_3]^-$ fragment to a polarized $[F_2]$ fragment. A stable linear configuration with $C_{\infty v}$ symmetry is observed at 2.6 kJ/mol above the global energy minimum. It is connected to an equivalent structure by a stationary point at the high symmetry point, which transforms according to the $D_{\infty h}$ point group.

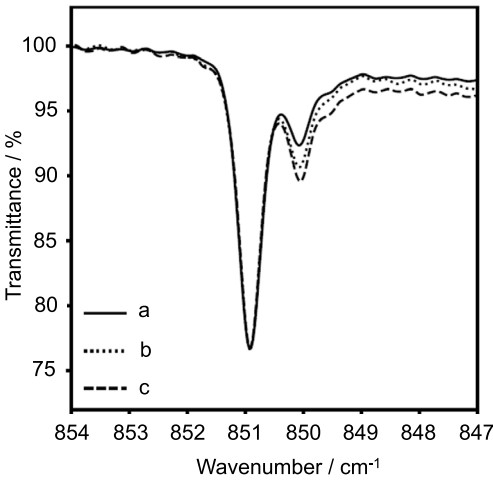
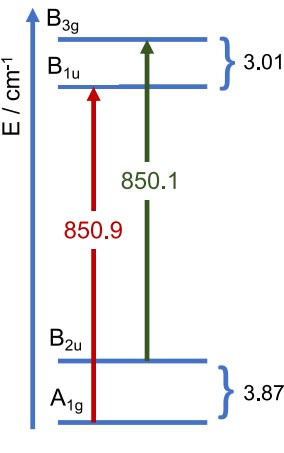

**Fig. 2 | FTIR spectra for [F₅]⁻.** The spectra (left panel) were obtained after co-deposition of potassium fluoride with 1% $F_2$ in solid neon at 5K. **a** After 3 h deposition (solid), **b** after 5 h irradiation using the IR source (dotted), **c** upon annealing to 10 K (dashed). All spectra were scaled to match the intensity of the dominant band in spectrum a). The right panel shows a diagram of the calculated vibrational energy levels (blue) and the symmetry allowed transitions (red and black) as observed in experiment (all energies in cm⁻¹). The states are labeled according to the $D_{2h}$ point group. Source data are provided as a Source Data file.

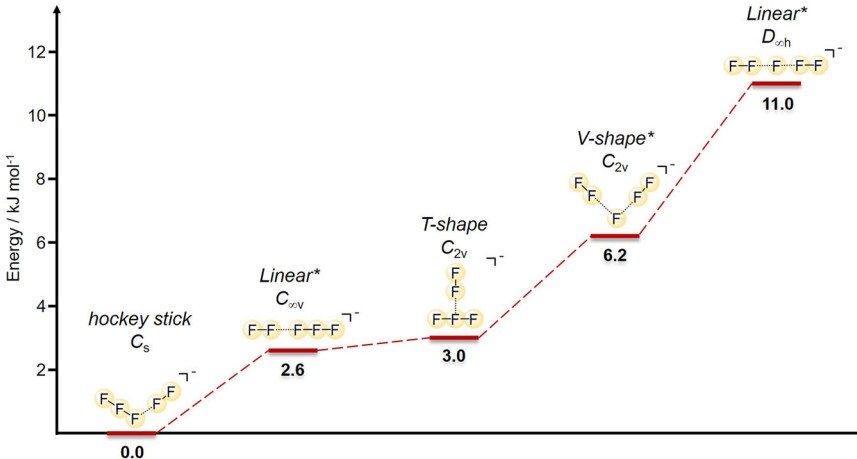

**Fig. 3 | The possible structures of [F₅]⁻ in gas phase.** Relative energies of the different stable structures and transition states of [F₅]⁻ in gas phase were obtained from CCSD(T)/aug-cc-pVQZ calculations. In gas phase, both linear structures and the V-shape (all three labelled with *) structure are transition states.

A local minimum in a T-shaped planar structures is also found at ~3 kJ/mol above the global ground state. Finally, a stationary V-shaped structure is found at 6.2 kJ/mol above the hockey stick (see Suppl. Fig. 1 for a graphical representation of how the different structures are connected).

To better understand the potential landscape and extract the important spectroscopic signatures in the gas phase, two models of the [F₅]⁻ anion are investigated: one containing planar isomers (hockey stick, V-shape, T-shape) around the global energy minimum, and one localized model restricted to linear isomers. Both models are parametrized using first principle electronic structure calculations and the potential energy surfaces (PES) are adjusted to adequate analytical forms, as described in the SM (see also Suppl. Tab. 2–5).

The lowest-energy structures of [F₅]⁻ in the planar model are shown in Fig. 4a). The isomers can be understood as a distorted $F_2$ dimer with a dangling F⁻ holding the structure together. The choice of diatom-diatom Jacobi coordinates depicted in Fig. 4b) offers enough flexibility to describe all isomers in the vicinity of the global potential energy minimum. The distance between the $F_2$ units is described by $R$ and the dangling F⁻ is at a distance $r_c$ of the center-of-mass of the $F_2$ dimer. The transformation between the isomers follows angle $\theta_t$, which hints at a tunnelling path between the left and right structures. Finally, the internal stretches of the $F_2$ fragments are represented by $\{r_a, r_b\}$. Two equivalent hockey stick isomers (labeled $C_s$) are connected by a low barrier of 6.2 kJ/mol, which corresponds to the V-shaped transition state. They can be identified in the 2D potential energy cut in the $R\theta_t$-plane depicted in the bottom panel of Fig. 4c).

The global minima $C_s$ are further connected by a small barrier (6.9 kJ/mol) via the rotation of the tunnelling angle to T-shaped local minima (see bottom two structures of Fig. 4a). As the tunnelling angle $\theta_t$ increases, the F⁻ ion migrates from left to right, thereby changing the charge distribution in the molecule. Analysis of the electronic structure reveals that the negative charge resides predominantly on terminal fluorine atoms of the [F₃]⁻ fragment up to the transition state. The structure can be interpreted as a continuous transformation from a [F₂]–[F₃]⁻ to a [F₃]⁻–[F₂] isomer. The transformation is also accompanied by a variation of the FF bond lengths from 1.446 Å in the polarized [F₂] fragment to 1.707 Å in the [F₃]⁻ fragment. The central atom is found at 2.278 Å and 1.762 Å from the nearest atom in the respective fragments. Upon transformation, the dipole moments of the molecule and within each fragment thereby change sign.

The ground vibrational state is found to be delocalized over both minima corresponding to the isomers transforming according to the $C_s$ point group. The zero-point energy is split into two components

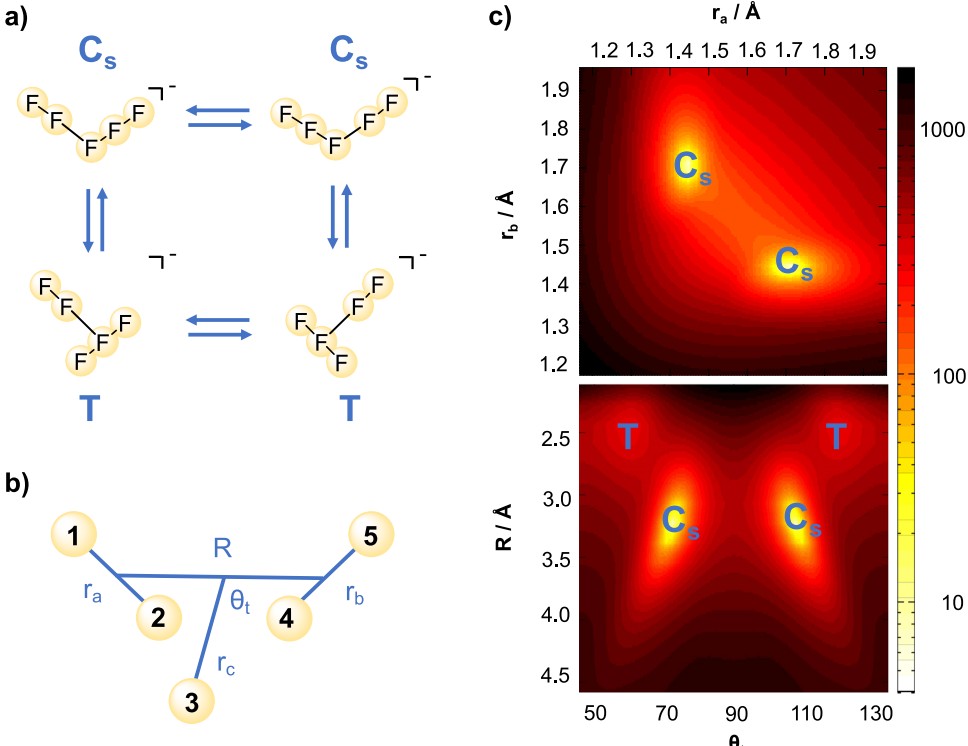

**Fig. 4 | Potential energy surface of [F$_5$]$^-$ in gas phase. a** Structure of the planar pentafluorine anion in its minimal C$_s$ isomer (top), and in T-shape isomers (bottom) at a relative energy of 3.0 kJ/mol, all connected by low-energy barriers. **b** Jacobi coordinates used to study the planar model: {$r_a$, $r_b$} are internal F$_2$ stretches connected via their center-of-mass by coordinate $R$, and $\theta_t$ is the angle describing the tunnelling motion of the central fluorine atom, at distance $r_c$ of the center-of-mass of the F$_2$ dimer. **c** Two-dimensional cuts of the potential energy surface along selected coordinates in the vicinity of the minimal energy isomer. Labels T and C$_s$ mark the positions of the T-shaped and hockey stick structures. Color map represents increasing energies from yellow to red and black. Source data are provided as a Source Data file.

that are symmetric and antisymmetric with respect to reflection about $\theta_t = 90°$. The overall structure transforms according to C$_{2v}$ point group because of quantum mechanical tunnelling between the two C$_s$ structures. Figure 5 shows 2D cuts of the nuclear probability density for the first excited states along the terminal F$_2$ bonds. Only the symmetric states are shown. The red and yellow arrows allow to correlate the nodal structure in the bottom panels with the nuclear displacements. The antisymmetric stretch of the [F$_3$]$^-$ fragment can be recognized in panel a), and panel b) shows the internal stretch of the [F$_2$] fragment. The latter bears significant density between the two C$_s$ minima, at $\theta_t = 75°$ and $\theta_t = 105°$ for $R = 3.3$ Å. Coupling between the two vibrating F$_2$ is also observed in the $r_a r_b$-plane. This leads to an energetic splitting of 4.81 cm$^{-1}$, which corresponds to 6.9 ps for tunnelling from one isomer to the other. The energetic splitting of the [F$_3$]$^-$ antisymmetric stretch (0.12 cm$^{-1}$) is much smaller, with a tunnelling rate around 1/277 ps$^{-1}$ much faster than the experimental resolution.

The agreement of the theoretical frequencies with experiment stems from the potential construction. Yet, the energy splitting calculated for the band around 850 cm$^{-1}$ is overestimated (4.8 vs -1 cm$^{-1}$ in experiment). Further, the transition dipole moments favor excitation of the [F$_3$]$^-$ antisymmetric stretch around 525 cm$^{-1}$. Indeed, the relative intensity is about 4:1 for the latter. The important discrepancy with experiment is that a single strong band is observed in the range 500–1000 cm$^{-1}$, while two bands are predicted theoretically: one strong signal for the [F$_3$]$^-$ antisymmetric stretch at -525 cm$^{-1}$ and a much weaker one for the [F$_2$] internal stretch at -850 cm$^{-1}$.

This conundrum can potentially be lifted considering the effect of the cryogenic matrix on the structure of the complex. In previous work, it was demonstrated that rare gas matrices can have important effects on vibrational frequencies of polyfluorine species[40–42]. Further,

the anion was found to occupy various cavities of different sizes and shapes, following different formation kinetics. We can speculate that a [F$_5$]$^-$ anion isolated in a neon matrix would most likely adopt a linear isomer based on steric considerations. A linear structure transforms according to a higher symmetry group and it could therefore possess IR-forbidden vibrational modes. In the gas phase, the linear [F$_5$]$^-$ anion is found marginally above the global minimum at 2.6 kJ/mol, an energy small enough to likely be compensated by the matrix. To investigate the linear pentafluorine anion a potential energy surface in internal coordinates was determined. Internal bond coordinates provide a more natural choice to represent the most important degrees of freedom of the linear pentafluorine anion. They are depicted in Fig. 6b), together with two dimensional cuts of the potential energy surface in panel a).

The potential energy reveals the presence of two equivalent minima transforming according to the C$_{\infty v}$. They correspond to a weakly bound linear complex in either the [F$_3$]$^-$–[F$_2$] or [F$_2$]–[F$_3$]$^-$ isomer. The potential minima are shown in Fig. 6a) and labeled L/R throughout. The conversion barrier is comparable to that found for the planar model. As in the planar case, the transfer of the F$^-$ anion from one F$_2$ to the other implies an equivalent redistribution of the charges in the complex and an inversion of the dipole moment along the molecular axis. The outer bond lengths ($r_1$ and $r_4$) are also affected, changing from 1.72 Å to 1.43 Å in [F$_3$]$^-$ and [F$_2$], respectively. Simultaneously, the inner bond lengths range from 1.75 Å–2.34 Å, with the transition state at $r_2 = r_3 = 2.04$ Å.

A tunnelling splitting of 2.88 cm$^{-1}$ in the zero-point energy corresponds to a transfer time of 11.6 ps. Two dominant contributions emerge in the spectrum. They can be attributed to the antisymmetric stretch of the [F$_3$]$^-$ fragment and to the internal stretch of the polarized

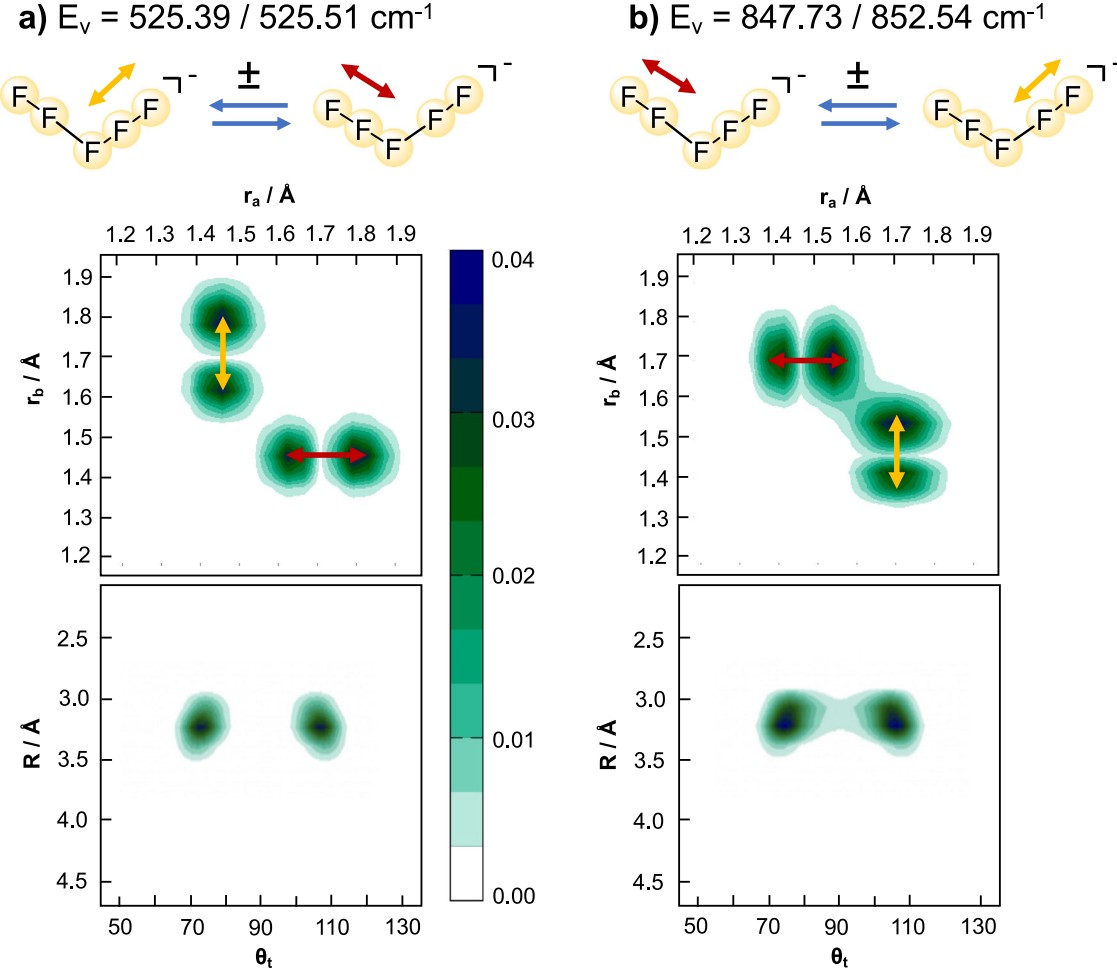

**Fig. 5 | Two-dimensional nuclear probability density cuts for selected modes involving the $\{r_a \pm r_b\}$ motion for the planar $[F_5]^-$ model.** The red/yellow arrows highlight the nuclear displacements. **a** Antisymmetric $[F_3]^-$ stretching mode, for which there is a node at $\theta_t = 90°$ and the probability density is vanishingly small between the lobes. **b** Internal $F_2$ stretching mode, where probability density is observed between the lobes for both in the $R\theta_t$- and $r_ar_b$-planes, indicative of tunnelling. Source data are provided as a Source Data file.

$[F_2]$ moiety. There is a significant red shift of both transitions as compared with the planar model. The antisymmetric $[F_3]^-$ stretching mode is more affected ($-96\ \text{cm}^{-1}$ and $-78\ \text{cm}^{-1}$ for the antisymmetric and symmetric states, respectively), as compared with the internal $[F_2]$ stretch ($-32\ \text{cm}^{-1}$ for the symmetric state).

The nuclear probability density of the two modes are shown in Fig. 7. Skewed displacements in the $r_1r_2$- and $r_3r_4$-planes indicate the presence of intermode coupling. Tunnelling is not directly observed in the latter vibrational eigenstate (see $r_2r_3$-plane in Fig. 7) but the presence of an energetic splitting of $17.7\ \text{cm}^{-1}$ between the antisymmetric and symmetric states is a reminder that the states indeed tunnel. On the contrary, the nuclear probability density for the internal $[F_2]$ vibration is delocalized in the $r_2r_3$-plane. As can be seen from the left panel of Fig. 7, the wave function covers both minima of the tunnelling $F^-$ ion, and some probability density is observed around the transition state ($r_2 = r_3 \approx 1.95$ Å). Evidence of tunnelling is also present in the $r_1r_3$- and $r_3r_4$-planes, where the nodal structure is not resolved because of the large probability density between the local minima.

Computing the transition intensities reveals that the low-energy transition at $447.65\ \text{cm}^{-1}$ has the largest intensity, and the band at $814.46\ \text{cm}^{-1}$ is vanishingly small. This is due to the behaviour of the latter state w.r.t. reflection about $\theta_t = 90°$, which is forbidden according to the dipole selection rule. Interestingly, the antisymmetric band $\{r_1 - r_4\}$ of the polarized $F_2$ molecule is not observed in the spectrum. This is in part because the state is found much above the tunnelling

barrier and its character becomes strongly mixed with other modes, preventing a unique assignment.

## Influence of the neon matrix

Previous theoretical investigations of $[F_3]^-$ in neon and argon matrices have revealed important effects of the environment on the spectroscopic signatures[40–42]. Neon was found to interact weakly with the anion and thus favor insertion inside a hexagonal cavity, independently of the number of vacancies in the matrix. This leads to a slightly compressed, linear configuration for $[F_3]^-$ and a blue shift in the spectrum consistent with experimental observations[18–20]. Three-body interactions were found to weaken the relative interaction of $[F_3]^-$ with more polarizable matrices such as argon, leading to the observed red shift compared to Ne matrices in the FTIR spectrum. Condensation kinetics was found to explain the preference of the linear $[F_3]^-$ to occupy a two-fold vacancy cavity[42]. These findings hint at an important contribution of matrix effects to understand the most stable configuration of $[F_5]^-$ in cryogenic neon, as the formation of $[F_5]^-$ is not experimentally observed in argon matrices or any other matrix host material.

The matrix is found to only marginally affect the potential energy surface of the planar model (see Suppl. Tab. 3–5 for details). To study the interaction of the Ne-matrix environment with the linear $[F_5]^-$ anion, the effect of the two neon atoms lying along the axis of the $[F_5]^-$ is considered (see Fig. 8a). The influence of the matrix is considered for fixed values of the cavity length ($d$). The size of the linear species in gas

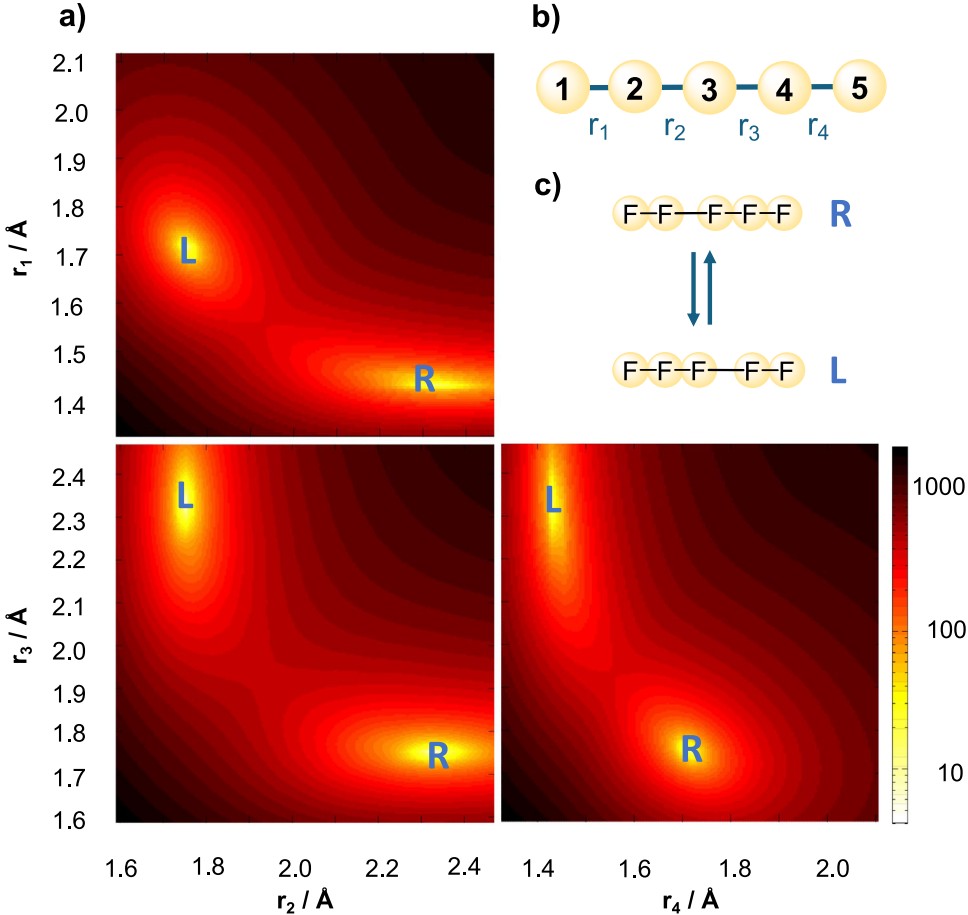

**Fig. 6 | Potential energy surface of linear [F₅]⁻ in gas phase. a** Two-dimensional cuts of the potential energy surface along selected coordinates for linear structures. **b** Internal coordinates used to investigate the linear model: {$r_1$, $r_2$, $r_3$, $r_4$} are the distances between neighbouring atoms. **c** The two linear [F₅]⁻ local minima in the C$_{\infty v}$ isomer are connected by an energy barrier of 8.43 kJ/mol. They are found at an energy of 2.6 kJ/mol above the global energy minimum. Color map represents increasing energies from yellow to red and black. Source data are provided as a Source Data file.

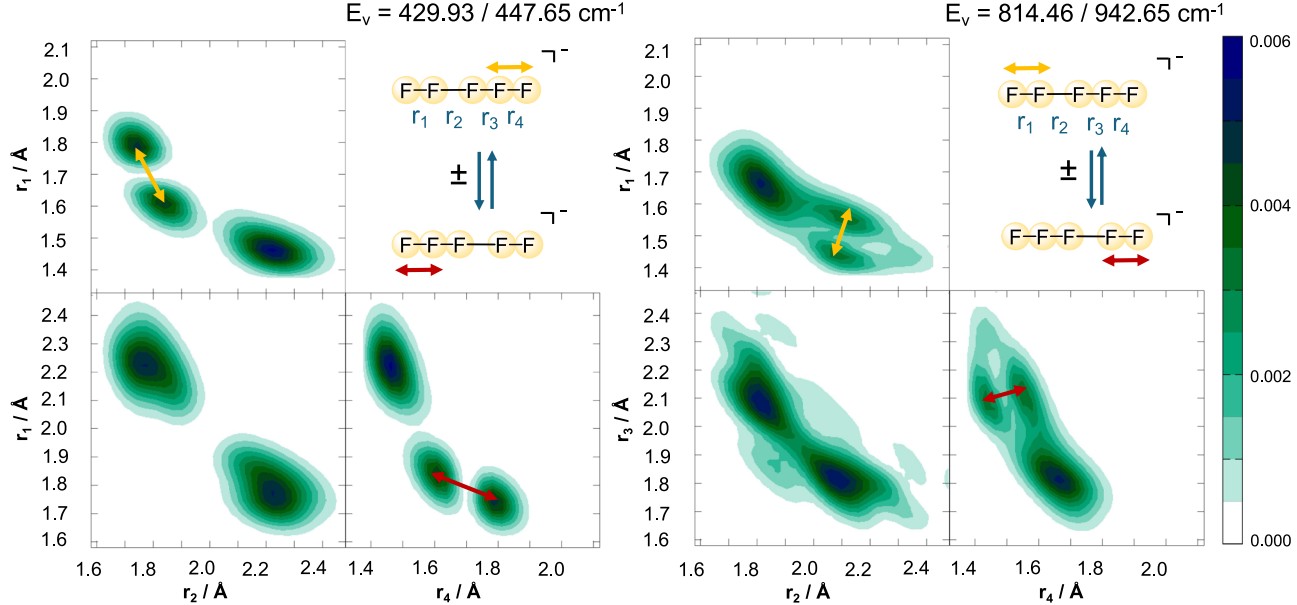

**Fig. 7 | Two-dimensional nuclear probability density cuts for the first (left) and second (right) IR-active internal modes {$r_1 \pm r_4$} of the linear [F₅]⁻ anion.** The red/ yellow arrows highlight that the nuclear displacements take place at the outer bonds within the [F₃]⁻ fragments. Source data are provided as a Source Data file.

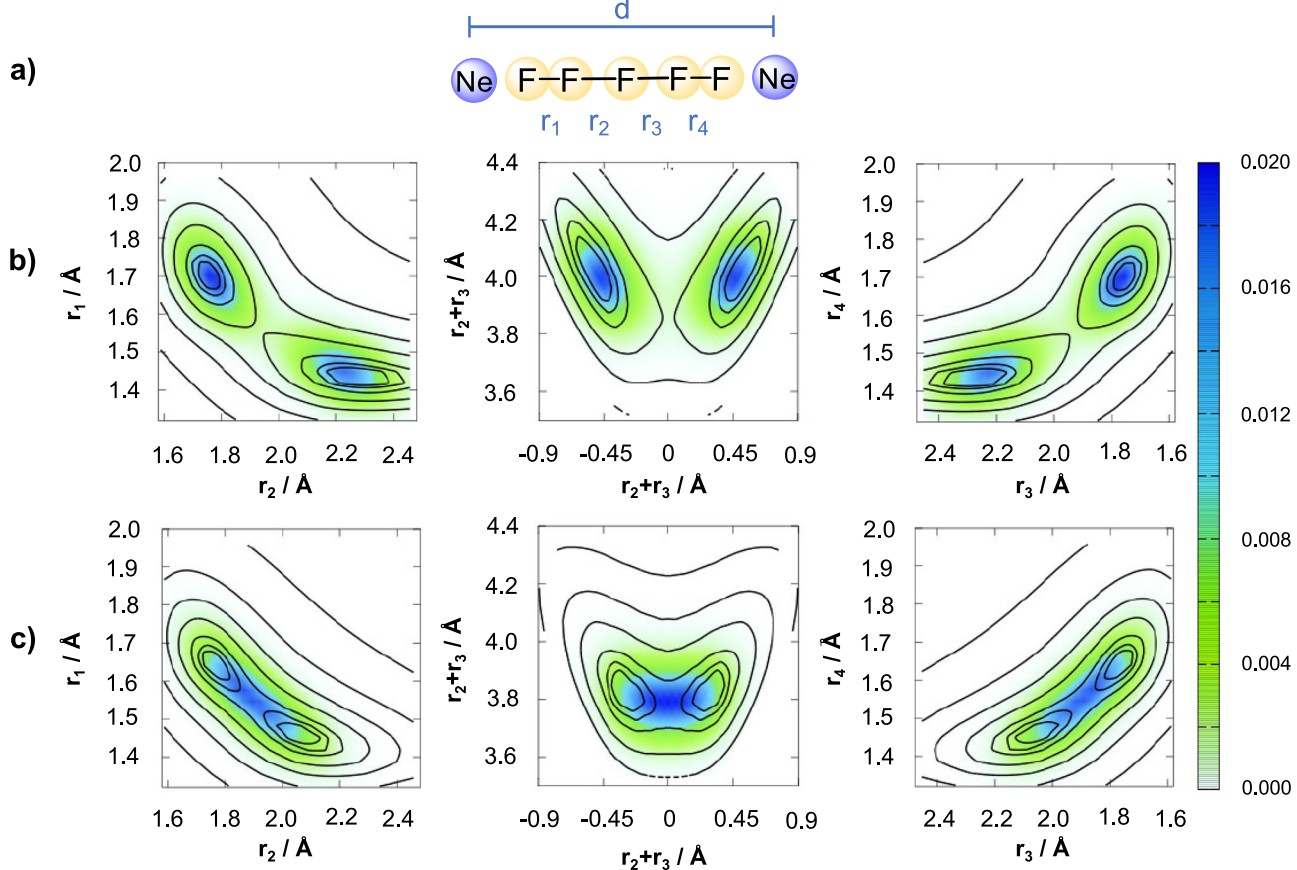

**Fig. 8 | Effect of a neon cavity on the energetic stability of a linear [F₅]⁻ anion. a** Structure of the [F₅Ne₂]⁻ cluster. The bond coordinates {$r_1$, $r_2$, $r_3$ $r_4$} and the cavity length $d$ are represented. Neon and fluorine atoms are depicted in ice blue and green, respectively. A color map of the ground vibrational state probability density superimposed on a contour representation of the potential energy surface (PES) is shown for a cavity length of (**b**) $d = 12.6$ Å, and (**c**) $d = 11.6$ Å. Contour lines for the PES at {20, 50, 100, 200, 500, 1000, 2000, 5000} cm⁻¹. Source data are provided as a Source Data file.

phase corresponds roughly to a triple vacancy in the solid Ne-lattice ( ~12.6 Å long). The electronic energy for a range of linear configurations is calculated at the coupled cluster level of theory with single, double, and perturbative triple excitations; CCSD(T)/aug-cc-pVTZ[43–46]. The associated potential energy corrections are computed at the MP3 level of theory[47] and interpolated using permutation invariant polynomials[48], as described in the SM.

Two-dimensional cuts of the resulting potential energy surface (PES) are shown as contours in Fig. 8b and c for two values of the cavity length. Color maps of the nuclear probability density of the vibrational ground state are overlaid. The PES forms a double-well which connects the [F₃-F₂]⁻ and [F₂-F₃]⁻ configurations via a low-energy barrier. As the cavity is compressed from $d = 12.6$ Å to $d = 11.6$ Å, stronger confinement in the matrix leads to a bond compression and the energetic stabilization of the [F₅]⁻ anion. By comparing the central panel of Fig. 8b, c, a reduction of the pentafluoride length by ~0.2 Å is observed. The nuclear probability density of the ground vibrational state also reveals that there is indeed tunnelling through the barrier, even in the largest cavity. The tunnelling barrier diminishes when the cavity becomes smaller because the transition state structure appears less affected by compression. At $d = 11.6$ Å, the tunnelling barrier is so small that the ground vibrational state becomes delocalized in a symmetric configuration that belongs to the $D_{\infty h}$ point group. The anion inside the cavity should thus be understood as a central fluorine anion shared between two polarized F₂ molecules.

**Quasi-linear anion inside a cavity**
From these findings, we construct a model potential for the tunnelling fluorine anion inside a compressed cavity. The electronic energy of the

[F₅]⁻ anion interacting with a neon cavity composed of the 18 closest neon atoms is computed from density functional theory (DFT) calculations[49,50] using the dispersion-corrected B3LYP-D3 functional as implemented in the Gaussian16 program[51] and an aug-cc-pVTZ basis[45,46] on all atoms. All 18 neon atoms of the cavity are included in DFT calculations and they are kept frozen during geometry optimization. It is found that the [F₅]⁻ anion adopts a slightly bent configuration inside the cavity. This quasi-linear structure can move almost freely inside the cavity, with a small rotation barrier observed when all fluorine atoms are far from the neon atoms of the matrix. The rotation of the quasi-linear structure involves all atoms of the pentafluoride, with the central F atom undergoing the motion with largest amplitude. As such, it is the tunnelling-defining atom[35].

Rotation inside the cluster shown in Fig. 9 leads to a double well potential (see Suppl. Fig. 2) and the associated barrier is found to vary strongly with the length of the cavity. It ranges from 32 and 3780 cm⁻¹ for cavity lengths between $d = [10.6, 12.6]$ Å, which are consistent with the CCSD(T) simulations of the linear Ne₂-[F₅]⁻ model described in the previous Section. The choice of functional strongly affects the rotational barrier, which varies by about an order of magnitude. This shortcoming of DFT is expected for a system containing many fluorine atoms which exhibits strong charge-shift character, that has to be described by valence bond methods or equally well with either multiconfiguration methods (e.g. MRCI) or sophisticated post-Hartree-Fock methods (e.g. CCSD(T)). The latter method was successfully used for the description of F₂ and [F₃]⁻ which exhibit similar problems when using DFT. To account for this limitation, we explore the different regimes for the rotational barrier obtained by DFT in the construction of a minimal spectroscopic model.

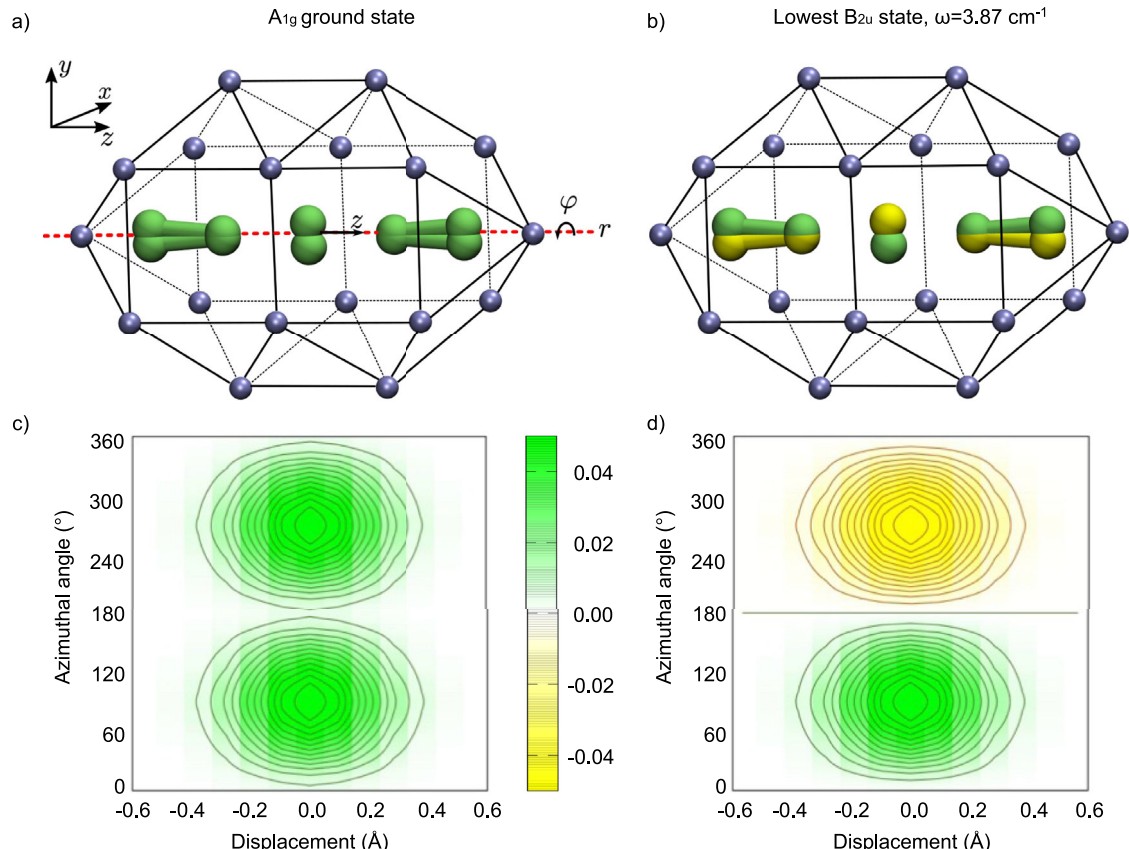

**Fig. 9 | Ground vibrational state of a quasi linear [F₅]⁻ inside a neon cavity.**
**a**, **b** Schematic representation of the configuration [F₅]⁻ vibrational motion in the totally symmetric $A_{1g}$ and antisymmetric state of symmetry $B_{2u}$. Positive and negative values of the vibrational wave function are shown in green and yellow, respectively. Neon atoms are depicted in ice blue. The dynamical coordinates $\{z, \varphi\}$ represent respectively the displacement of the central atom along, and the rotation of the molecule around the precession axis $r$, shown as a dashed red line. **c**, **d** Associated color maps for the vibrational state probability density. Contour lines equally spaced at $0.005 a_0^{-2}$. Source data are provided as a Source Data file.

Hirshfeld charge analysis reveals that the central fluorine atom ($-0.33e$) as well as the two terminal atoms ($-0.20e$) bear significant negative charge in the symmetric configuration, while the two other atoms are almost neutral ($-0.07e$). We found that bending out of the cavity planes does not affect the charge distribution, while displacement of the central atom along the molecular axis leads to a large variation of the dipole moment. This is a strong indication that this mode is likely to be active in IR absorption spectroscopy. A minimal spectroscopic model should thus also include the IR-active displacement of the central F atom from its equilibrium position, $z$, as well as the azimuthal angle $\varphi$ describing the frustrated rotation of the slightly bent [F₅]⁻ anion around the precession axis $r$ inside the cavity (dashed red line in Fig. 9). The Hamiltonian of the model system takes the form

$$\hat{H}(z, \varphi) = -\frac{\hbar^2}{2m_F}\frac{\partial^2}{\partial z^2} - \frac{\hbar^2}{2\mathcal{I}}\frac{\partial^2}{\partial \varphi^2} + V(z, \varphi) \quad (1)$$

The potential $V(z, \varphi) = V_0\cos^n[\varphi] + V_1 z^2 + V_c z^2 \sin^2[2\varphi]$ respects the symmetry of the [F₅]⁻ precession movement inside the cavity. A value of $n = 20$ reproduces approximately the width of the rotational barriers, which corresponds to a rotation by 60° inside the cavity. For different values of the frustrated rotation barrier, we optimize the harmonic force constant ($V_1$), the coupling strength ($V_c$), and the moment of inertia $\mathcal{I}$ to reproduce the experimental data (see Suppl. Tab. 1). Upon optimization, the model is found to reproduce the ground state splitting of 3.87 cm⁻¹ and the position of the two observed IR-active bands at 850.9 cm⁻¹ and 850.1 cm⁻¹ exactly in all cases.

Independently of the choice of rotational barrier within the range obtained from DFT calculations, the vibrational states shown in Figs. 9 and 10 take the same general form. In the totally symmetric ground vibrational state, the central atom is seen tunnelling above and below the $xz$-plane of the flattened cavity. This naturally leads to a splitting of the ground state, resulting in an antisymmetric state of symmetry $B_{2u}$ at 3.87 cm⁻¹ (see Figs. 2 and 9b). This state behaves like a bending mode out of the plane of the cavity in which the rotational barrier is highest. For the different rotational barriers, the main change is found in the value of the moment of inertia of [F₅]⁻, which varies from 277 $m_e$ Å² to 1200 $m_e$ Å² and 3751 $m_e$ Å² with decreasing barrier height ($V_0 = 3500$, 350, and 35 cm⁻¹, respectively). The latter value compares well with the moment of inertia obtained from DFT, which is 1968 $m_e$ Å² for a cavity of length 11.6 Å. By approximating that all F−F bond lengths are equal, the displacement of the central atom perpendicular to the precession axis, $r$, can be evaluated from the relation $\mathcal{I} \simeq (3/2)m_F r^2$. This displacement is found to range from 0.07 Å to 0.27 Å for the largest to the smallest barriers studied, in line with the DFT result ($r = 0.24$ Å for cavity of length 11.6 Å). This is a strong indication that the [F₅]⁻ anion remains almost linear inside the cavity.

Figure 10 shows the lowest-lying IR-active vibrational states obtained by anharmonic variational calculations (c.f. SM for methodological details). The excited states are accessible either from the the $A_{1g}$ vibrational ground state (transition from Fig. 9a–10a), or the lowest state of $B_{2u}$ symmetry (from Figs. 9b–10b). It can be seen that the excited states (Figs. 9d and 10d) develop a node in the $xy$-plane compared to their respective initial state (Figs. 9c, 10c), while retaining the fingerprints of the tunnelling potential along $\varphi$. Neglecting intermode

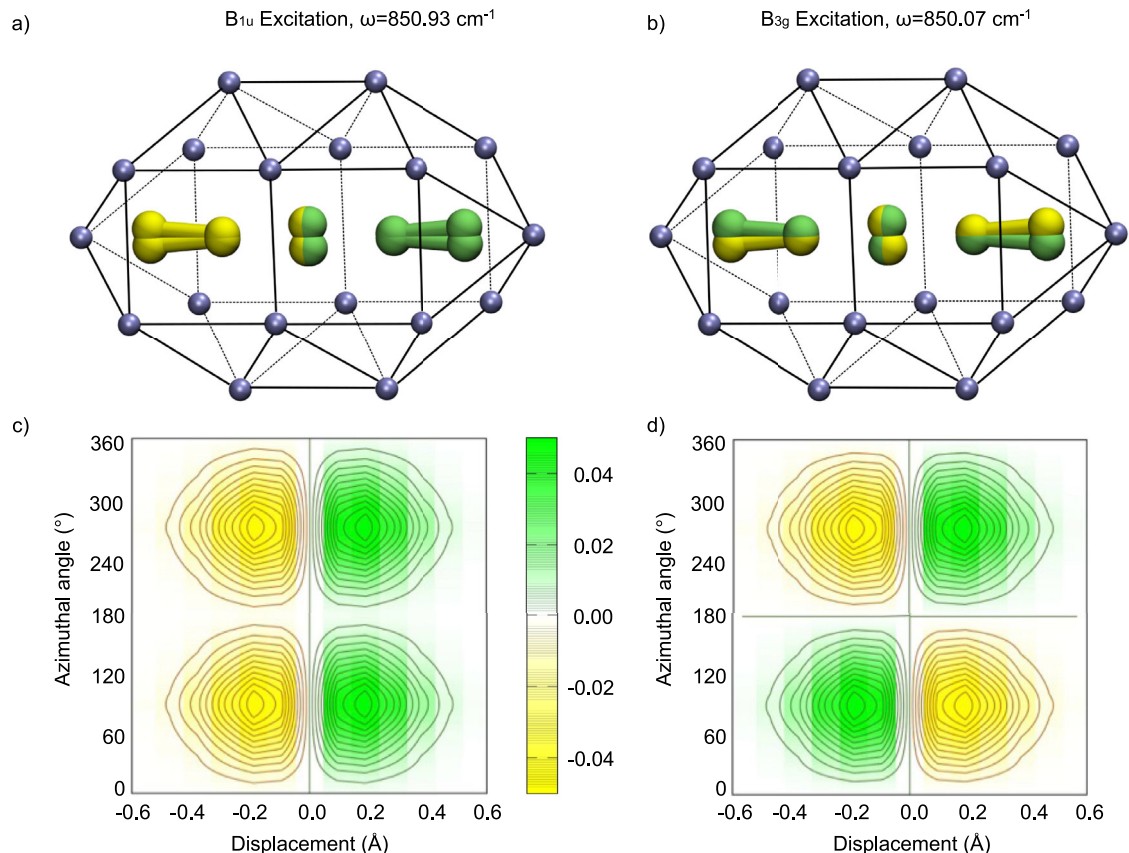

**Fig. 10 | Lowest IR-active vibrational states of a quasi linear [F₅]⁻ inside a neon cavity. a, b** Schematic representation of the configuration [F₅]⁻ vibrational motion in the totally symmetric $A_{1g}$ and antisymmetric state of symmetry $B_{2u}$. Positive and negative values of the vibrational wave function are shown in green and yellow, respectively. Neon atoms are depicted in ice blue. **c, d** Associated color maps for the vibrational state probability density. Contour lines equally spaced at $0.005a_0^{-2}$. Source data are provided as a Source Data file.

coupling yields two transitions at the same frequency. The values of $V_c$ range between 10–20% of the harmonic frequency for small to intermediate potential barriers, which describe a weakly coupled system. For the largest potential barrier investigated ($V_0$ = 3500 cm⁻¹), the coupling becomes about 80% of the harmonic force constant. Despite this strong coupling, the system vibrational wave functions and energies remain similar as in the low barrier/weak coupling limit. The adaptability of the model on such a large range of rotational barriers show strong evidence that tunnelling inside the neon cavity dictates the IR-activity of [F₅]⁻.

## Discussion

In this work, we present a joint experimental/theoretical study of the spectroscopic signature of configuration [F₅]⁻ in a cryogenic neon matrix. Our simulations yield convincing theoretical evidence for heavy-atom QMT in a polyfluoride species with the central fluorine as the tunnelling-determining atom. Calculation of the coupled, anharmonic vibrational eigenstates demonstrate the presence of tunnelling in the pentafluorine anion in gas phase. An important energetic splitting of the ground vibrational states is observed, with the time for tunnelling from one isomer to the other ranging from ~10–360 ps in the linear and planar models, respectively. On the other hand, the computed transition intensities from the ground state predict a strong signal associated with the [F₃]⁻ unit in the [F₅]⁻ configuration, inconsistent with the experimental observations. These findings hint at an important influence of the neon matrix, which is found to stabilize the [F₅]⁻ anion in a symmetric, linear configuration. A model is constructed that is compatible with the available experimental observations. It suggests that the strong IR signal observed for the Ne-matrix isolated

[F₅]⁻ is due to a tunnelling [F]⁻ anion within a F₂/Ne cavity, and the splitting of this signal confirms a tunnelling motion of this central fluorine atom – the heaviest tunnelling-determining atom involved in QMT known to date. This study demonstrates that confinement in a weakly interacting matrix environment can induce QMT in higher polyfluoride compounds.

## Methods

### Experimental details

Potassium fluoride (KF) targets were produced from dry KF powder using a hydraulic lab press. Filling of the pressing die and removal of the KF targets were carried out under an argon atmosphere inside a glove box in order to prevent contact of the highly hygroscopic material with atmospheric water. Matrix-isolation experiments were carried out in a custom-built vacuum chamber equipped with a cold-head, a helium compressor unit, and a gilded copper plate (matrix support) mounted on the cold-head. The device was held at a temperature of 5 K during experiments. Over the course of an experiment the vacuum chamber was kept at pressures between $3 \cdot 10^{-7}$ and $1 \cdot 10^{-6}$ mbar using an oil diffusion pump. For laser-ablation, pulses of a Nd:YAG laser (1064 nm) were aligned through a hole in the matrix support and focused on the potassium fluoride target. The KF targets were mounted on a target holder and rotated using a magnetically coupled electric motor. A gas inlet next to the target holder connected to a gas line via a needle valve allowed for the controlled co-deposition of gas mixtures with laser-ablated material. In a typical experiment potassium fluoride was evaporated using laser pulses at a rate of 1 Hz and co-deposited with Ne/F₂ (99:1) gas mixtures at 5 K. The deposits were accumulated over the course of 3–4 hours at flow rates between

1–1.4 mbar L min$^{-1}$. After deposition, the matrix support was rotated by 90° to measure IR spectra in reflection mode. IR spectra with a resolution of $\leq 0.1$ cm$^{-1}$ were recorded using a Bruker Vertex 80v vacuum FTIR spectrometer. A transfer optic was used to guide the IR measurement beam to the sample and to align the beam reflected by the matrix support to the detector of the instrument. Annealing of the deposits was facilitated by a heater cartridge built into the cold head. LEDs of 273 and 730 nm wavelengths were used for irradiation of the deposits.

### Computational details

For computing the potential energy surfaces, electronic structure calculations of gas phase [$F_5$]$^-$ are performed using coupled cluster with single, double, and perturbative triple excitations, CCSD(T)[43,44]. For the planar model, all atoms are described using an aug-cc-pVQZ basis set[45,46], while the linear model utilizes an aug-cc-pVTZ basis set. The dipole moment is determined at the MP2/aug-cc-pVTZ level of theory[47]. The wave function-based electronic structure calculations were carried out using the MOLPRO program package[39].

For the planar model (see Fig. 4b), the potential energy is sampled uniformly around the global minimum, with further sampling points around linear configurations. The electronic structure calculations are performed by optimizing the remaining coordinates under $R\theta_t$ constraints. The internal $F_2$ bond is sampled around the local optimal bond length. The dataset is symmetrized by reflection about $\theta_t = 90°$. The data is fitted using the Levenberg-Marquardt algorithm, as implemented in Gnuplot. For the linear model, symmetry unique configurations are sampled around all stable and stationary points of the potential energy surface, which is fitted using the monomial symmetrization approach[48] A dipole moment surface is obtained by the same approach using points calculated at the MP2/aug-cc-pVTZ level of theory[47].

The guest-host interaction is accounted for as an additive correction to the potential energy. For the linear configurations, the guest-host interaction potential is modelled as the energy difference with and without two terminal Ne atoms around the linear [$F_5$]$^-$ configurations, computed at the MP3/aug-cc-pVTZ level of theory[47]. Finally, realistic parameters for the quasi-linear model in the neon cavity are computed at the density functional theory (DFT)[49,50]. The dispersion-corrected B3LYP-D3 functional implemented in Gaussian16[51] is used throughout. All atoms are represented using an aug-cc-pVTZ basis[45,46].

The vibrational Hamiltonians for the different models share the following general form

$$\hat{H} = \hat{T}_{\text{kin}} + V(q_1, q_2, \ldots, q_n) \tag{2}$$

where $\hat{T}_{\text{kin}}$ is the kinetic energy operator and $V(q_1, q_2, \ldots, q_n)$ is the potential energy along coordinates $q_1, q_2, \ldots, q_n$. We represent the vibrational wave functions using a direct product basis of one-dimensional discrete variable representation (DVR) functions

$$\psi_n(q_1, q_2, \ldots, q_n) = \sum_{\alpha_1 \alpha_2 \ldots \alpha_n} c^{(n)}_{\alpha_1 \alpha_2 \ldots \alpha_n} \chi_{\alpha_1}(q_1) \chi_{\alpha_2}(q_2) \cdots \chi_{\alpha_n}(q_n) \tag{3}$$

The resulting matrix representation is used to compute the low-lying vibrational eigenstates reported in the manuscript (see SM for technical details). For the quasi linear model, the potential parameters in Eq.(1) are varied through a python script and minimization is performed using the SciPy implementation of the Nelder-Mead method.

### Reporting summary

Further information on research design is available in the Nature Portfolio Reporting Summary linked to this article.

### Data availability

Source data are provided with this paper. The raw data for Figures 1–2 and 4–10 are provided with this paper. The cartesian coordinates for Fig. 3 are provided within the Supplementary Information (see Suppl. Tab. 6–7). Further data supporting this manuscript is available from the authors upon request. Source data are provided with this paper.

### Code availability

Molpro is a commercially available quantum chemical program, which was used for all quantum chemical calculations of [$F_5$]$^-$ and the matrix effects. Gaussian16 is a commercially available quantum chemistry program, which was used for all density functional theory calculations of [$F_5$]$^-$Ne$_x$. The figures were generated using the freely-available command-line driven graphing utility Gnuplot, the Python library Matplotlib, and the molecular visualization program VMD. IR spectra were collected using the Bruker Opus software and plotted using OriginPro and the Matplotlib Python package. Affinity Designer was used to combine figures and improve the readability of fonts. The custom programs explicitly tailored to study vibrations in this system are made available without warranty. Further instructions and support can be obtained from JCT upon request. The electronic structure data generated in this study are provided in the Supplementary Information/Source Data file.

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

## Acknowledgements

Funded by the Deutsche Forschungsgemeinschaft (DFG, German Research Foundation) – Project-ID 387284271 – SFB 1349 (CM, FR, FB, LC and SR), and the Elsa Neumann Stiftung des Landes Berlin (FB). Computational resources and support were granted through the high performance computing (HPC) group of Freie Universität's IT-service (FUB-IT). The authors are grateful to PD Dr. Denis Usvyat (Humboldt University Berlin) and Dr. Thomas Grohmann for stimulating discussions.

## Author contributions

C.M., H.B., and S.R. designed the project. F.A.R and H.B. carried out the matrix-isolation experiments. C.M. and J.C.T. developed the theoretical models. C.M., F.B., L.C., and J.C.T. performed quantum chemical calculations. B.P., S.R., and J.C.T. directed and coordinated the research. F.A.R., C.M., and J.C.T. wrote the manuscript. H.B., B.P., C.M., S.R., and J.C.T. revised the manuscript.

## Funding

## Competing interests

The authors declare no competing interests.
