## [Transparent Peer Review file · Nature Communications]

Experimental Observation of Quantum Mechanical Fluorine Tunnelling

Corresponding Author: Professor Sebastian Riedel

Version 0:

Reviewer comments:

Reviewer #1

(Remarks to the Author)

This is an excellent paper on an enigmatic species where the researchers show that they have learned quite a bit about the properties this new F5(-) entity. I have some technical comments as outlined as follows:

Fluoride tunneling has been implied in "Enhanced reactivity of fluorine with para-hydrogen in cold interstellar clouds by resonance-induced quantum tunnelling." Nat. Chem. 2019, 11, 744. Non-carbon heavy atom tunneling has also experimentally observed in "Isomerizations between Nitrosyl Halides X-N=O and Isonitrosyl Halides X-O-N: A Matrix-Spectroscopic Study." Chem. Eur. J. 2000, 6 (5), 800. In the broader context of the relative importance of QMT the authors may want to consider including "Quantum Mechanical Tunneling Is Essential to Understanding Chemical Reactivity." Trends Chem. 2020, 2, 980.

Theory: The use of at least four different levels [CCSD(T) with two basis sets, MP2, MP3, DFT...] of theory is hard to understand. For instance, MP3 is not at all a useful method for energy computations. These can also be done at the CCSD(T) level of theory. I understand why the dipole moments were computed at MP2 but it is clear that these are only approximations for the derivatives needed to get CCSD(T) IR intensities. For such a small structure one could also have used finite differences for the key IR frequencies at the CCSD(T) level. I understand that this all is a considerable effort but so are the experiments.

Was a damping function use (like BJ) for the D3 correction?

Tunneling phenomena are more typically expressed in half-lives, not absolute tunneling times. There is quite a debate about the meaning of tunneling time (from zero to measurable), so one should perhaps avoid this.

Nomenclature: conformations are related through single-bond rotations, so this is not quite the case for (-)F3---F2 going to F2---F3(-). These formally are degenerate minima that are in equilibrium.

Language: There are some typos such as "polyfluoride" and the paper should be checked carefully for these. "...heating of the deposit to 10K" is not incorrect but sounds rather funny; this can be expressed better.

Reviewer #2

(Remarks to the Author)

Heavy atom tunneling is an interesting phenomenon which was studied extensively in the last decade. Here, the authors report on the experimental and theoretical evidence for fluorine atom tunneling, which was postulated but never experimentally observed before. I find this an important extension of the field of nuclear quantum effects in general and would say that the study deserves publication in a general journal such as Nat.Comm. For me the work is conducted in a sound and consistent manner and the results support the claim of F atom tunneling.

I recommend publishing the work after the following minor concerns are responded to

1. I have one request regarding the cavity: Intuitively, including more atoms would lengthen the distance between the minima in normalized (e.g. mass-weighted Cartesian) coordinates. This would lead to a reduced tunneling probability as the barrier becomes broader. Could the authors compute the minimum-energy paths of the model they used in this study and compare it to a minimum-energy path of a model where all Ne atoms are included explicitly? As this is a mere test for the effect of including a cavity, also DFT level of theory be sufficient, e.g. B3LYP-D3/aug-cc-pVTZ as the authors have already used.
2. For me, it is not clear where the data in Figure 2b are coming from: Are these computations done in this study or are they literature values?
3. Figure 3: The authors state "Relative energies of the different stable structures and transition states[...]", but it is not clear which structures are minima and which ones are transition state structures (TSs). It becomes clear after carefully reading, but the accessibility of the work would benefit from adding more clarity here. Further more, they should state which structures are connected by which TS in order to make the potential energy surface more understandable.
4. The authors talked about the electronic structure, and used a single-reference method. Could they report on possible multi-reference-character here e.g. by performing MCSCF single points on saddle point structures. A higher level of accuracy than CCSD(T)/VQZ will probably not be achievable anyways, but it would help to learn about the electronic structure (or be sure about the closed-shell case).

Reviewer #3

(Remarks to the Author)

This manuscript by Müller et al. describes matrix isolation IR experiments and computational modeling on the [F5]⁻ anion, and makes that case that tunneling occurs in this species. Numerous researchers worldwide are probing the limits of quantum tunneling by heavy atoms; until now, oxygen is the heaviest element for which there is experimental evidence for tunneling. The current work revises that by showing, convincingly, that fluorine atoms tunnel. Thus, this report appears to be at the forefront of the field, and should be of interest to a broad readership. The IR spectroscopic evidence presented is convincing in showing bands due to the [F5]⁻ anion; the thermal behavior of the relevant peaks is consistent with tunneling combined with low-lying vibrational excited states, and inconsistent with the notion that matrix effects cause different peaks. The authors also report extensive computational results, in which they give thorough consideration to many different possible geometries of the species under consideration. They use appropriate levels of theory, including coupled cluster theory with a large basis set. In addition to gas-phase computations, they include neon cages to simulate the matrix, and with the latter case they are able to model the system in a way that fits the experimental IR results, including the observed tunneling splitting. I confess that the modeling/fitting techniques, including the details in the Supplementary Material, are beyond my understanding, so hopefully another reviewer can comment on those. For the rest of the manuscript, the conclusions seem well supported by the experimental and computational data.

My two small issues are:

1. On p. 6, the first sentence of the paragraph is unclear. "The ground vibrational state is found to be delocalized in both Cs." I do not understand what the authors are saying. Perhaps some clarification would help.
2. On p. 7, the authors mention Fig. 6c, but I do not see a panel c) in Fig. 6.

If those issues are addressed, then I recommend publication in Nature Communications.

Reviewer #4

(Remarks to the Author)

This manuscript details a comprehensive experimental and computational investigation into the structure of the F5⁻ anion assumed in neon matrices and its dynamic nature as facilitated by heavy-atom tunneling of the central fluorine atom. It serves as a remarkable report for not only the polyhalogenide and heavy-atom tunneling communities, but also for readers performing the matrix isolation spectroscopy and computational studies in the solid state in general with its diligent and comprehensive analyses. It is thus recommended for publication after minor revisions as outlined below.

Recommended minor changes

Abstract: "Quantum mechanical tunnelling occurs when an atom overcomes a classically forbidden potential energy barrier" should be re-phrased since it is unclear what "classically forbidden" means. It is suggested to use something along the lines of "Quantum mechanical tunnelling occurs when an atom penetrates a potential energy barrier that cannot be overcome thermally under the reaction conditions".

Abstract: "To date, it has been observed experimentally for all elements up to oxygen" – this is a strong statement that does not align itself with my knowledge of heavy-atom tunneling reactions. Would the others mind sharing the articles that demonstrate lithium tunneling? Beryllium tunneling is also questionable since the only instances of experimental observation of molecules containing beryllium that undergo tunneling rearrangements (to my knowledge) are cited later in the article as an example of oxygen tunneling (refs 28 and 29).

In general, the article in its abstract and the final paragraph of its introduction strongly stresses the idea of assigning (heavy-atom) tunneling reactions to specific elements, likely the atoms that move the largest distance during the reaction and/or whose KIE have been measured in the respective experimental studies (despite not all atoms being equally positioned for KIE experiments). While a common approximation delineates between hydrogen- and heavy-atom tunneling, it should be understood that quantum-mechanical tunneling always involves all atoms in the molecule (since its wavefunction,

describing the motion of all atoms, tunnels). A re-phrasing of the respective sections in the abstract and introduction is thus recommended. However, I would like to stress that this comment in no way subtracts from this manuscript's great achievement since obviously here fluorine is indeed the tunneling atom whose motion dominates the tunneling rearrangement (coined "tunneling-determining atom by Kozuch et al.; <https://pubs.rsc.org/en/content/articlelanding/2014/cp/c4cp00115j/unauth>) and since it may, according to Nandi, Kozuch and Kästner, be the heaviest atom to be capable to assume this role ("fluorine wall", <https://www.sciencedirect.com/science/article/abs/pii/S0009261420305935>).

Page 2: The extended discussions of the potential minimum energy structures of F5⁻ anion and the transition states connecting them is hard to follow and is marred by imprecise language, i.e. "transforming according to the Cs point group" and later "transforming according to the C2v point group" make it unclear whether these point groups are assigned to the minima or transition states. The confusion could be addressed by re-phrasing as well as explicit reference to Figs. 3 and 4 that provide unambiguous visualizations of the respective structures.

Page 2: Ref. 24 should be complemented by an earlier comprehensive review (that was later extended by a few updates in Ref. 24; <https://wires.onlinelibrary.wiley.com/doi/10.1002/wcms.1235>) and a later perspective (<https://chemistry-europe.onlinelibrary.wiley.com/doi/full/10.1002/chem.202201775>) that again contains updates upon Ref. 24.

Page 2: The short mention of ref. 31 at the end of the introduction should likely also point out its discussion of the tunneling rearrangement of IF6⁻ anion given its relevance to the topic of polyhalogenide ions discussed prior.

Page 4: It may be valuable to point out for the (potentially uninformed) reader that matrix site annealing is an irreversible process, further strengthening the authors' argument that the experimental observations indicate reversible thermal population of low-lying vibrational energy levels.

Fig. 5 caption: "Antisymmetric [F3]- stretching mode, for which tunnelling is not observed" could be changed to "for which tunneling cannot be observed due to the limited resolution of the experimental setup" to better resonate with the discussion in the main text.

Page 12: "Fig. 10 shows the lowest-lying IR-active vibrational states obtained by anharmonic variational calculations [refs. 45-49]" – it is unclear what purpose the citation of five references here serves, the latter four of which share an author with one of the corresponding authors of this manuscript. It is likely that this fragment stems from an earlier draft since it may point readers to refs. elucidating the methodology as it still does in the Supporting Material ("The low-lying eigenstates of the vibrational Hamiltonians are computed using a spectral-transform Lanczos eigensolver [refs. 11-14, identical to refs. 46-49]"). It is strongly recommended to delete the refs. here to dispel any incorrect, yet unfortunate impression of potential gratuitous self-citation that the reader may get from refs. 46-49 (which are otherwise not cited) and to substitute them by a reference to the Supporting Material (e.g. "cf. SM").

Supporting Material: No Cartesian coordinates of any of the computed structures (ground-state and transition states, with and without addition of neon atoms) can be found here which severely hinders the (otherwise straight-forward) reproducibility of the computational data. Have they been uploaded to a data depository instead?

Version 1:

Reviewer comments:

Reviewer #1

(Remarks to the Author)

The authors have addressed all issues I raised. I have no further queries.

Reviewer #2

(Remarks to the Author)

The authors addressed all points raised by the other reviewers and myself and I recommend publication of this nice piece of work.

Reviewer #4

(Remarks to the Author)

My compliments to the authors for such a rigorous implementation of the many comments by the reviewers. Revising this manuscript with these many comments in mind (some thankfully overlapping) cannot have been easy, but I consider the end result a great improvement and a valuable addition to the heavy-atom tunneling and polyhalogenide literature!

Reviewer #1 (Remarks to the Author):

This is an excellent paper on an enigmatic species where the researchers show that they have learned quite a bit about the properties this new F_5^- entity. I have some technical comments as outlined as follows:

Fluoride tunneling has been implied in *Enhanced reactivity of fluorine with para-hydrogen in cold interstellar clouds by resonance-induced quantum tunnelling*, *Nat. Chem.* **2019**, *11*, 744. Non-carbon heavy atom tunneling has also experimentally observed in *Isomerizations between Nitrosyl Halides $X-N=O$ and Isonitrosyl Halides $X-O-N$: A Matrix-Spectroscopic Study*, *Chem. Eur. J.* **2000**, *6*(5), 800. In the broader context of the relative importance of QMT the authors may want to consider including *Quantum Mechanical Tunneling Is Essential to Understanding Chemical Reactivity*, *Trends Chem.* **2020**, *2*, 980.

These are excellent suggestions, and we have added the references in the revised version of the manuscript.

Theory: The use of at least four different levels [CCSD(T) with two basis sets, MP2, MP3, DFT...] of theory is hard to understand. For instance, MP3 is not at all a useful method for energy computations. These can also be done at the CCSD(T) level of theory.

This is an interesting remark. Various levels of theory have been used for the different scenarios investigated. In each case, we have used the best level of theory affordable that would provide an answer to our scientific questions. The first part of the manuscript deals with two models of the F_5^- in gas phase, for which CCSD(T) was used throughout. The number of points required to calculate high-dimensional potential energy surfaces (PES) is large. The planar model requires constrained geometry optimization for each point to keep the number of dimensions low. We have thus used a smaller basis than for the linear model, for which the dimensionality is reduced by imposing that all atoms are on a line.

For the same reason, we cannot afford to compute the guest-host interaction energy at the CCSD(T) level of theory. We have thus used MP2 and MP3 levels of theory to simulate the guest-host interaction as an energy difference between embedded and free-standing F_5^- for all structures required for the construction of the PES. These perturbative methods scale as $O(N^5)$ and $O(N^6)$, compared to $O(N^7)$ for CCSD(T), leading to huge numerical savings. We have confirmed that the energy difference between the minimum and the transition state is accurate up to 5 meV when comparing the MP3 and CCSD(T) levels (see SM). This is because MP3 is the cheapest numerical method that includes three-body electron correlation, which apparently significantly contributes to the guest-matrix interaction.

Finally, DFT is only used in the last part of the manuscript, which is concerned with the modeling of F_5^- inside the matrix. DFT was used to simulate a quasi-linear F_5^- in the matrix including 18 neon atoms. Due to the complexity of the F_5^- electronic structure, DFT is not expected to be accurate, since it fails already for the stationary points of the molecule's PES. This is why we use DFT only to define the range for the structural parameters of the model for F_5^- in the matrix that would be physically reasonable. The model is then optimized within this physically-reasonable range, providing convincing evidence of the working hypothesis.

I understand why the dipole moments were computed at MP2 but it is clear that these are only approximations for the derivatives needed to get CCSD(T) IR intensities. For such a small structure one could also have used finite differences for the key IR frequencies at the CCSD(T) level. I understand that this all is a considerable effort but so are the experiments.

Indeed, structure optimizations at the CCSD(T) level are very expensive. As the referee mentioned, no 1st or 2nd order energy derivatives are available at this level. In the simplest case – a $\text{Ne}_2[\text{F}_5]^-$ in $C_{\infty v}$ symmetry – calculating the forces along the main axis in each structure optimization step would require 15 single point calculations for up to 14 (twice the number of coordinates) steps. The Hessian matrix of this system has $(7 \times (7-1)/2 + 7 =)$ 28 unique elements associated with the coordinate axis along the molecule's main axis, requiring 112 single-point calculations. In total, determining vibrational frequencies would require about 320 single-point calculations. However, the numerically accurate result would suffer from the shortcomings of a simple embedding model for the matrix consisting of just two Ne atoms. Instead, we fitted a potential energy surface to structures of an isolated $[\text{F}_5]^-$ calculated at the CCSD(T) and corrected for the effect of two Ne-atoms representing the matrix and evaluated at MP3 level (see comment above). We believe that the vibrational frequencies obtained analytically from this potential energy surface are in no way less accurate than those we could have calculated directly with the approach outlined above.

Was a damping function used (like BJ) for the D3 correction?

All DFT-D3 calculations were performed with the Gaussian program applying the damping function as originally proposed by Chai and Head-Gordon (*Phys. Chem. Chem. Phys.* **2008**, *10*(44), 6615) and implemented by Grimme *et al.* (*J. Chem. Phys.* **2010**, *132*, 154104) for their D2 version of the empirical dispersion correction.

Tunnelling phenomena are more typically expressed in half-lives, not absolute tunnelling times. There is quite a debate about the meaning of tunneling time (from zero to measurable), so one should perhaps avoid this.

This is an interesting debate. A half-life would refer to tunnelling through a barrier describing a first-order process, such as the decay of a resonance or an asymmetric (and potentially irreversible) reaction. In the present case, tunneling leads to the transformation from one stable structure to another equivalent one. The process is completely reversible and the population dynamics oscillate periodically between the two structures. It is the period of this half-oscillation that we report as tunneling time. It is directly related to the energy splitting between symmetric and anti-symmetric vibrational ground states, which are delocalized on both structures. In a realistic system, dephasing will destroy the coherences in the system and suppress tunneling. We thus understand tunnelling as “tunnelling on the timescale of the experiment”. To avoid controversy, we have modified the three occurrences where we could find “tunnelling time” in the manuscript.

Nomenclature: conformations are related through single-bond rotations, so this is not quite the case for $(-)\text{F}_3\text{---F}_2$ going to $\text{F}_2\text{---F}_3(-)$. These formally are degenerate minima that are in equilibrium.

We have changed nomenclature to “isomers” to avoid confusion with the chemical definition of “conformations”. Indeed, these are two equivalent structures related by quantum tunnelling, similar to two isomers that readily interconvert. Since the latter are called “tautomers”, a term such as “quantum tautomers” or “quantomers” would fit very well. But we prefer to avoid introducing new terminology.

Language: There are some typos such as "polyfloride" and the paper should be checked carefully for these. "...heating of the deposit to 10K" is not incorrect but sounds rather funny; this can be expressed better.

We have corrected all typos we could find and modified the text concerning the heating of the deposit.

Reviewer #2 (Remarks to the Author):

Heavy atom tunneling is an interesting phenomenon which was studied extensively in the last decade. Here, the authors report on the experimental and theoretical evidence for fluorine atom tunneling, which was postulated but never experimentally observed before. I find this an important extension of the field of nuclear quantum effects in general and would say that the study deserves publication in a general journal such as Nat.Comm. For me the work is conducted in a sound and consistent manner and the results support the claim of F atom tunneling.

I recommend publishing the work after the following minor concerns are responded to

1. I have one request regarding the cavity: Intuitively, including more atoms would lengthen the distance between the minima in normalized (e.g. mass-weighted Cartesian) coordinates. This would lead to a reduced tunneling probability as the barrier becomes broader. Could the authors compute the minimum-energy paths of the model they used in this study and compare it to a minimum-energy path of a model where all Ne atoms are included explicitly? As this is a mere test for the effect of including a cavity, also DFT level of theory be sufficient, e.g. B3LYP-D3/aug-cc-pVTZ as the authors have already used.

This is precisely how we have treated the effect of the cavity in the last part of the manuscript, where we include all 18 neon lying closest to the F_5^- . We have clarified the text in the main manuscript.

2. For me, it is not clear where the data in Figure 2b are coming from: Are these computations done in this study or are they literature values?

Indeed, the vibrational energies of the states and the corresponding transitions shown in the right panel of Figure 2 are results from our calculations. We have modified the caption to clarify this.

Old caption

Fig. 2 FTIR spectra obtained after co-deposition of potassium fluoride with 1% F_2 in solid neon at 5K. a) After 3 h deposition (solid), b) after 5 h irradiation using the IR source (dotted), c) upon annealing to 10 K (dashed). All spectra were scaled to match the intensity of the dominant band in spectrum a). The right panel shows a diagram of the energy levels and the observed transitions (all energies in cm^{-1}). The states are labeled according to the D2h point group.

New caption

Fig. 2 FTIR spectra (left panel) obtained after co-deposition of potassium fluoride with 1% F_2 in solid neon at 5K. a) After 3 h deposition (solid), b) after 5 h irradiation using the IR source (dotted), c) upon annealing to 10 K (dashed). All spectra were scaled to match the intensity of the dominant band in spectrum a). The right panel shows a diagram of the calculated vibrational energy levels and the symmetry allowed transitions as observed in experiment (all energies in cm^{-1}). The states are labeled according to the D_{2h} point group.

3. Figure 3: The authors state "Relative energies of the different stable structures and transition states[...]", but it is not clear which structures are minima and which ones are transition state structures (TSs). It becomes clear after carefully reading, but the accessibility of the work would benefit from adding more clarity here.

We have altered Figure 3, indicating which structures are minima (*hockey stick* and *T-shape*) and which are transition states, and we have included this information in the caption of Figure 3 as well.

Furthermore, they should state which structures are connected by which TS in order to make the potential energy surface more understandable.

We have included a new figure with a graph in the SI, showing which structures are connected.

Figure S1: Graphical representation of how the different structures (cf. Figure 3) of $[F_5]^-$ are connected. Transition state structures are marked with an asterisk. The numbers in parenthesis give the relative energy of the structures in kJ/mol.

4. The authors talked about the electronic structure, and used a single-reference method. Could they report on possible multi-reference-character here e.g. by performing MCSCF single points on saddle point structures. A higher level of accuracy than CCSD(T)/VQZ will

probably not be achievable anyways, but it would help to learn about the electronic structure (or be sure about the closed-shell case).

Indeed, systems with fluorine-fluorine bonds (and a large number of lone-pairs) are known to show substantial correlation effects due to exceptionally strong charge shift effects in these bonds. However, from previous experience with $[\text{F}_3]^-$ we believe that calculations at CCSD(T) level are sufficiently accurate for $[\text{F}_5]^-$ as well. In our coupled cluster calculations of the linear symmetric isomer of $[\text{F}_5]^-$, which in our belief should show the most multi-reference character of all isomers – the t1-diagnostic – a commonly accepted measure for multi-reference character – yields a value of 0.023. This is only very slightly above the commonly accepted threshold of 0.02, below which a system is considered single reference. For systems with significant multi-reference character, the value of the t1 diagnostic would be much higher and as Alavi et al. pointed out (*J. Chem. Theory Comput.* **2018**, *14*, 6240), diagnostics based on singles amplitudes are at best indirect indicators of multi-reference character which might still be of no problem for CCSD(T).

In addition, we now performed CASSCF(36,20) calculations for the $[\text{F}_5]^-$ in $D_{\infty h}$ symmetry. Since the first excited state lies about 0.20 Hartree above the ground state, we conclude that multi-reference character in this system is negligible. We have added a short description of these calculations in the SM.

Reviewer #3 (Remarks to the Author):

This manuscript by Müller et al. describes matrix isolation IR experiments and computational modeling on the $[\text{F}_5]^-$ anion, and makes that case that tunneling occurs in this species. Numerous researchers worldwide are probing the limits of quantum tunneling by heavy atoms; until now, oxygen is the heaviest element for which there is experimental evidence for tunneling. The current work revises that by showing, convincingly, that fluorine atoms tunnel. Thus, this report appears to be at the forefront of the field, and should be of interest to a broad readership. The IR spectroscopic evidence presented is convincing in showing bands due to the $[\text{F}_5]^-$ anion; the thermal behavior of the relevant peaks is consistent with tunneling combined with low-lying vibrational excited states, and inconsistent with the notion that matrix effects cause different peaks. The authors also report extensive computational results, in which they give thorough consideration to many different possible geometries of the species under consideration. They use appropriate levels of theory, including coupled cluster theory with a large basis set. In addition to gas-phase computations, they include neon cages to simulate the matrix, and with the latter case they are able to model the system in a way that fits the experimental IR results, including the observed tunneling splitting. I confess that the modeling/fitting techniques, including the details in the Supplementary Material, are beyond my understanding, so hopefully another reviewer can comment on those. For the rest of the manuscript, the conclusions seem well supported by the experimental and computational data.

My two small issues are:

1. On p. 6, the first sentence of the paragraph is unclear. "The ground vibrational state is found to be delocalized in both Cs." I do not understand what the authors are saying. Perhaps some clarification would help.

We have modified the sentence to make it clearer that the ground state is delocalized and could be understood as a superposition of two states localized in the two minima labeled "Cs" of the potential energy surface.

2. On p. 7, the authors mention Fig. 6c, but I do not see a panel c) in Fig. 6.

We thank the referee for noticing the mistake, which we have corrected in the revised version of the manuscript.

If those issues are addressed, then I recommend publication in Nature Communications.

Reviewer #4 (Remarks to the Author):

This manuscript details a comprehensive experimental and computational investigation into the structure of the F_5^- anion assumed in neon matrices and its dynamic nature as facilitated by heavy-atom tunneling of the central fluorine atom. It serves as a remarkable report for not only the polyhalogenide and heavy-atom tunneling communities, but also for readers performing the matrix isolation spectroscopy and computational studies in the solid state in general with its diligent and comprehensive analyses. It is thus recommended for publication after minor revisions as outlined below.

Recommended minor changes

Abstract: "Quantum mechanical tunnelling occurs when an atom overcomes a classically forbidden potential energy barrier" should be re-phrased since it is unclear what "classically forbidden" means. It is suggested to use something along the lines of "Quantum mechanical tunnelling occurs when an atom penetrates a potential energy barrier that cannot be overcome thermally under the reaction conditions".

This is an excellent comment. We have reformulated the abstract as per the referee's suggestion: „when a molecules goes from one state to another, between which there is a potential energy barrier that cannot be overcome thermally under the reaction conditions”

Abstract: "To date, it has been observed experimentally for all elements up to oxygen" – this is a strong statement that does not align itself with my knowledge of heavy-atom tunneling reactions. Would the others mind sharing the articles that demonstrate lithium tunneling? Beryllium tunneling is also questionable since the only instances of experimental observation of molecules containing beryllium that undergo tunneling rearrangements (to my knowledge) are cited later in the article as an example of oxygen tunneling (refs 28 and 29).

The referee is right. Tunnelling has been observed for elements up to oxygen but not for all elements up to oxygen.

In general, the article in its abstract and the final paragraph of its introduction strongly stresses the idea of assigning (heavy-atom) tunneling reactions to specific elements, likely the atoms that move the largest distance during the reaction and/or whose KIE have been

measured in the respective experimental studies (despite not all atoms being equally positioned for KIE experiments). While a common approximation delineates between hydrogen- and heavy-atom tunneling, it should be understood that quantum-mechanical tunneling always involves all atoms in the molecule (since its wavefunction, describing the motion of all atoms, tunnels). A re-phrasing of the respective sections in the abstract and introduction is thus recommended. However, I would like to stress that this comment in no way subtracts from this manuscript's great achievement since obviously here fluorine is indeed the tunneling atom whose motion dominates the tunneling rearrangement (coined "tunneling-determining atom" by Kozuch et al.; <https://pubs.rsc.org/en/content/articlelanding/2014/cp/c4cp00115j/unauth>) and since it may, according to Nandi, Kozuch and Kästner, be the heaviest atom to be capable to assume this role ("fluorine wall", <https://www.sciencedirect.com/science/article/abs/pii/S0009261420305935>).

We thank the referee for her/his appreciation of our work. In the revised version of the manuscript, we have reformulated the text in terms of tunnelling-determining atom, which is a more accurate description of the collective motion required for tunnelling. We have also added in the introduction a few words concerning the fluorine wall, which now begs to be broken under the appropriate reaction conditions that might be accessible via matrix-isolation spectroscopy.

Page 2: The extended discussions of the potential minimum energy structures of F_5^- anion and the transition states connecting them is hard to follow and is marred by imprecise language, i.e. "transforming according to the C_s point group" and later "transforming according to the C_{2v} point group" make it unclear whether these point groups are assigned to the minima or transition states. The confusion could be addressed by re-phrasing as well as explicit reference to Figs. 3 and 4 that provide unambiguous visualizations of the respective structures.

Stating that a structure/isomer "transforms according" to e.g. the C_{2v} point group does not refer to any transition of the structure. Instead, it is the mathematically accurate way of saying that this structure/isomer "shows C_{2v} symmetry". While the former expression is well known to everyone acquainted with group theory, the latter colloquial expression might only be understandable to a small group of chemists. Hence, we will not change this in the manuscript.

Page 2: Ref. 24 should be complemented by an earlier comprehensive review (that was later extended by a few updates in Ref. 24; <https://wires.onlinelibrary.wiley.com/doi/10.1002/wcms.1235>) and a later perspective (<https://chemistry-europe.onlinelibrary.wiley.com/doi/full/10.1002/chem.202201775>) that again contains updates upon Ref. 24.

We thank the referee for these references on carbon QMT, which we have added to the revised version of the manuscript.

Page 2: The short mention of ref. 31 at the end of the introduction should likely also point out its discussion of the tunneling rearrangement of IF_6^- anion given its relevance to the topic of polyhalogenide ions discussed prior.

We have modified the introduction to mention tunneling rearrangement of IF_6^- as well, as it is indeed relevant to the present topic.

Page 4: It may be valuable to point out for the (potentially uninformed) reader that matrix site annealing is an irreversible process, further strengthening the authors' argument that the experimental observations indicate reversible thermal population of low-lying vibrational energy levels.

We have altered the text at the end of chapter 2 to stress the irreversibility of matrix site annealing.

old text at the end of chapter 2:

This is indicative of a tunnelling behaviour rather than to the presence of matrix sites. Matrix sites are bands that are shifted with respect to the main band due to different molecular orientations within their cavity. These typically show the opposite thermal behavior and merge with the main band upon annealing, because softening of the matrix leads to reorientation of these molecules.

new text:

This is indicative of a tunnelling behaviour rather than to the presence of matrix sites. Matrix sites give rise to bands that are shifted with respect to the main band due to different molecular orientations within their cavity. These typically show the opposite thermal behavior and merge irreversibly with the main band upon annealing, because softening of the matrix leads to reorientation of these molecules.

Fig. 5 caption: "Antisymmetric $[\text{F}_3]^-$ stretching mode, for which tunnelling is not observed" could be changed to "for which tunneling cannot be observed due to the limited resolution of the experimental setup" to better resonate with the discussion in the main text.

This is a simple misunderstanding due to poor phrasing on our part. The figure shows no probability density between the lobes in the asymmetric state because of the node at $\theta = 90^\circ$. The symmetric state has a large probability between the lobes because of strong tunnelling. We have corrected the figure caption accordingly.

Page 12: "Fig. 10 shows the lowest-lying IR-active vibrational states obtained by anharmonic variational calculations [refs. 45-49]" – it is unclear what purpose the citation of five references here serves, the latter four of which share an author with one of the corresponding authors of this manuscript. It is likely that this fragment stems from an earlier draft since it may point readers to refs. elucidating the methodology as it still does in the Supporting Material ("The low-lying eigenstates of the vibrational Hamiltonians are computed using a spectral-transform Lanczos eigensolver [refs. 11-14, identical to refs. 46-49]"). It is strongly recommended to delete the refs. here to dispel any incorrect, yet unfortunate impression of potential gratuitous self-citation that the reader may get from refs. 46-49 (which are otherwise not cited) and to substitute them by a reference to the Supporting Material (e.g. "cf. SM").

Since the methodological details have been moved to the Supporting Material in the submitted version, we have replaced the citations by "cf. SM for methodological details", which should be sufficient to ensure reproducibility of the calculations by a trained theoretical vibrational spectroscopist.

Supporting Material: No Cartesian coordinates of any of the computed structures (ground-state and transition states, with and without addition of neon atoms) can be found here which severely hinders the (otherwise straight-forward) reproducibility of the computational data. Have they been uploaded to a data depository instead?

All coordinate files for the sampling of the potential energy surfaces – more than a few thousands – cannot be included in the SM. Nonetheless, we have added the cartesian coordinates for all gas phase structures reported in Fig.1, for the DFT calculations of the of the rotational barriers, as well as a typical structure for the matrix embedding of the linear molecule. These structures, together with the description of the PES sampling in the SM, should ensure reproducibility of the computational data.

hockey-stick, C_{2v}, 0.0 kJ/mol

9	0.0000000000	-1.4002846927	-0.2060800192
9	0.0000000000	-0.0669558058	1.6404162454
9	0.0000000000	-0.2111727172	-1.5061319642
9	0.0000000000	0.7398863244	2.8401626305
9	0.0000000000	0.9385268913	-2.7683668925

linear, C_{∞v}, 2.6 kJ/mol

9	0.0000000000	0.0000000000	-1.9229636953
9	0.0000000000	0.0000000000	-3.6314968140
9	0.0000000000	0.0000000000	2.1396172486
9	0.0000000000	0.0000000000	-0.1651057592
9	0.0000000000	0.0000000000	3.5799490199

T-shape, C_{2v}, 3.0 kJ/mol

9	0.0000000000	0.0000000000	-0.0044099199
9	0.0000000000	0.0000000000	2.4655447309
9	0.0000000000	0.0000000000	3.8941171902
9	0.0000000000	-1.7395929200	0.0023739994
9	0.0000000000	1.7395929200	0.0023739994

V-shape, C_{2v}, 6.2 kJ/mol

9	0.0000000000	0.0000000000	-1.2247470328
9	0.0000000000	1.5555475136	-0.1335202836
9	0.0000000000	-1.5555475136	-0.1335202836
9	0.0000000000	2.8631196490	0.7458937999
9	0.0000000000	-2.8631196490	0.7458937999

linear, D_{∞h}, 11.0 kJ/mol

9	0.0000000000	0.0000000000	-0.4816120561
9	0.0000000000	0.0000000000	1.0777247058
9	0.0000000000	0.0000000000	3.0000000000
9	0.0000000000	0.0000000000	4.9222752942
9	0.0000000000	0.0000000000	6.4816120561